# ON NEURAL NETWORK GENERALIZATION VIA PROMOTING WITHIN-LAYER ACTIVATION DIVERSITY

## ABSTRACT

During the last decade, neural networks have been intensively used to tackle various problems and they have often led to state-of-the-art results. These networks are composed of multiple jointly optimized layers arranged in a hierarchical structure. At each layer, the aim is to learn to extract hidden patterns needed to solve the problem at hand and forward it to the next layers. In the standard form, a neural network is trained with gradient-based optimization, where the errors are back-propagated from the last layer back to the first one. Thus at each optimization step, neurons at a given layer receive feedback from neurons belonging to higher layers of the hierarchy. In this paper, we propose to complement this traditional 'between-layer' feedback with additional 'within-layer' feedback to encourage diversity of the activations within the same layer. To this end, we measure the pairwise similarity between the outputs of the neurons and use it to model the layer's overall diversity. By penalizing similarities and promoting diversity, we encourage each neuron to learn a distinctive representation and, thus, to enrich the data representation learned within the layer and to increase the total capacity of the model. We theoretically study how the within-layer activation diversity affects the generalization performance of a neural network in a supervised context and we prove that increasing the diversity of hidden activations reduces the estimation error. In addition to the theoretical guarantees, we present an empirical study confirming that the proposed approach enhances the performance of neural networks.

## 1 INTRODUCTION

Neural networks are a powerful class of non-linear function approximators that have been successfully used to tackle a wide range of problems. They have enabled breakthroughs in many tasks, such as image classification (Krizhevsky et al., 2012), speech recognition (Hinton et al., 2012a), and anomaly detection (Golan & El-Yaniv, 2018). Formally, the output of a neural network consisting of P layers can be defined as follows:

$$f(\boldsymbol{x}; \mathbf{W}) = \phi^P(\boldsymbol{W}^P(\phi^{P-1}(\cdots \phi^2(\boldsymbol{W}^2\phi^1(\boldsymbol{W}^1\boldsymbol{x}))))), \tag{1}$$

where $\phi^i(.)$ is the element-wise activation function, e.g., *ReLU* and *Sigmoid*, of the $i^{th}$ layer and $\mathbf{W} = \{\boldsymbol{W}^1, \dots, \boldsymbol{W}^P\}$ are the corresponding weights of the network. The parameters of $f(\boldsymbol{x}; \mathbf{W})$ are optimized by minimizing the empirical loss:

$$\hat{L}(f) = \frac{1}{N} \sum_{i=1}^{N} l\big(f(\boldsymbol{x}_i; \mathbf{W}), y_i\big), \tag{2}$$

where $l(\cdot)$ is the loss function, and $\{\boldsymbol{x}_i, y_i\}_{i=1}^{N}$ are the training samples and their associated ground-truth labels. The loss is minimized using the gradient decent-based optimization coupled with back-propagation.

However, neural networks are often over-parameterized, i.e., have more parameters than data. As a result, they tend to overfit to the training samples and not generalize well on unseen examples (Goodfellow et al., 2016). While research on Double descent (Belkin et al., 2019; Advani et al., 2020; Nakkiran et al., 2020) shows that over-parameterization does not necessarily lead to overfitting, avoiding overfitting has been extensively studied (Neyshabur et al., 2018; Nagarajan & Kolter,

2019; Poggio et al., 2017) and various approaches and strategies have been proposed, such as data augmentation (Goodfellow et al., 2016), regularization (Kukačka et al., 2017; Bietti et al., 2019; Arora et al., 2019), and dropout (Hinton et al., 2012b; Wang et al., 2019; Lee et al., 2019; Li et al., 2016), to close the gap between the empirical loss and the expected loss.

Diversity of learners is widely known to be important in ensemble learning (Li et al., 2012; Yu et al., 2011) and, particularly in deep learning context, diversity of information extracted by the network neurons has been recognized as a viable way to improve generalization (Xie et al., 2017a; 2015b). In most cases, these efforts have focused on making the set of weights more diverse (Yang et al.; Malkin & Bilmes, 2009). However, diversity of the activation has not received much attention.

Inspired by the motivation of dropout to co-adapt neuron activation, Cogswell et al. (2016) proposed to regularize the activations of the network. An additional loss using cross-covariance of hidden activations was proposed, which encourages the neurons to learn diverse or non-redundant representations. The proposed approach, known as Decov, has empirically been proven to alleviate overfitting and to improve the generalization ability of neural network, yet a theoretical analysis to prove this has so far been lacking.

In this work, we propose a novel approach to encourage activation diversity within the same layer. We propose complementing 'between-layer' feedback with additional 'within-layer' feedback to penalize similarities between neurons on the same layer. Thus, we encourage each neuron to learn a distinctive representation and to enrich the data representation learned within each layer. Moreover, inspired by Xie et al. (2015b), we provide a theoretical analysis showing that the within-layer activation diversity boosts the generalization performance of neural networks and reduces overfitting.

Our contributions in this paper are as follows:

- Methodologically, we propose a new approach to encourage the 'diversification' of the layer-wise feature maps' outputs in neural networks. The proposed approach has three variants based on how the global diversity is defined. The main intuition is that by promoting the within-layer activation diversity, neurons within the same layer learn distinct patterns and, thus, increase the overall capacity of the model.

- Theoretically, we analyse the effect the within-layer activation diversity on the generalization error bound of neural network. The analysis is presented in Section 3. As shown in Theorems 3.7, 3.8, 3.9, 3.10, 3.11, and 3.12, we express the upper-bound of the estimation error as a function of the diversity factor. Thus, we provide theoretical evidence that the within-layer activation diversity can help reduce the generalization error.

- Empirically, we show that the within-layer activation diversity boosts the performance of neural networks. Experimental results show that the proposed approach outperforms the competing methods.

## 2 WITHIN-LAYER ACTIVATION DIVERSITY

We propose a diversification strategy, where we encourage neurons within a layer to activate in a mutually different manner, i.e., to capture different patterns. To this end, we propose an additional within-layer loss which penalizes the neurons that activate similarly. The loss function $\hat{L}(f)$ defined in equation 2 is augmented as follows:

$$\hat{L}_{aug}(f) = \hat{L}(f) + \lambda \sum_{i=1}^{P} J^i, \tag{3}$$

where $J^i$ expresses the overall pair-wise similarity of the neurons within the $i^{th}$ layer and $\lambda$ is the penalty coefficient for the diversity loss. As in (Cogswell et al., 2016), our proposed diversity loss can be applied to a single layer or multiple layers in a network. For simplicity, let us focus on a single layer.

Let $\phi_n^i(\boldsymbol{x}_j)$ and $\phi_m^i(\boldsymbol{x}_j)$ be the outputs of the $n^{th}$ and $m^{th}$ neurons in the $i^{th}$ layer for the same input sample $\boldsymbol{x}_j$. The similarity $s_{nm}$ between the the $n^{th}$ and $m^{th}$ neurons can be obtained as the average similarity measure of their outputs for $N$ input samples. We use the radial basis function to

express the similarity:

$$s_{nm} = \frac{1}{N} \sum_{j=1}^{N} \exp\big( - \gamma ||\phi_n^i(\boldsymbol{x}_j) - \phi_m^i(\boldsymbol{x}_j)||^2 \big), \tag{4}$$

where $\gamma$ is a hyper-parameter. The similarity $s_{nm}$ can be computed over the whole dataset or batch-wise. Intuitively, if two neurons $n$ and $m$ have similar outputs for many samples, their corresponding similarity $s_{nm}$ will be high. Otherwise, their similarity $s_{mn}$ is small and they are considered "diverse". Based on these pair-wise similarities, we propose three variants for the global diversity loss $J^i$ of the $i^{th}$ layer:

- **Direct:** $J^i = \sum_{n \neq m} s_{nm}$. In this variant, we model the global layer similarity directly as the sum of the pairwise similarities between the neurons. By minimizing their sum, we encourage the neurons to learn different representations.

- **Det:** $J^i = -\det(\mathbf{S})$, where $\boldsymbol{S}$ is a similarity matrix defined as $\boldsymbol{S}_{nm} = s_{nm}$. This variant is inspired by the Determinantal Point Process (DPP) (Kulesza & Taskar, 2010; 2012), as the determinant of $\boldsymbol{S}$ measures the global diversity of the set. Geometrically, $\det(\boldsymbol{S})$ is the volume of the parallelepiped formed by vectors in the feature space associated with $s$. Vectors that result in a larger volume are considered to be more "diverse". Thus, maximizing $\det(\cdot)$ (minimizing $-\det(\cdot)$) encourages the diversity of the learned features.

- **Logdet:** $J^i = -\text{logdet}(\mathbf{S})^1$. This variant has the same motivation as the second one. We use logdet instead of det as logdet is a convex function over the positive definite matrix space.

It should be noted here that the first proposed variant, i.e., direct, similar to Decov (Cogswell et al., 2016), captures only the pairwise diversity between components and is unable to capture the higher-order "diversity", whereas the other two variants consider the global similarity and are able to measure diversity in a more global manner.

Our newly proposed loss function defined in equation 3 has two terms. The first term is the classic loss function. It computes the loss with respect to the ground-truth. In the back-propagation, this feedback is back-propagated from the last layer to the first layer of the network. Thus, it can be considered as a between-layer feedback, whereas the second term is computed within a layer. From equation 3, we can see that our proposed approach can be interpreted as a regularization scheme. However, regularization in deep learning is usually applied directly on the parameters, i.e., weights (Goodfellow et al., 2016; Kukačka et al., 2017), while in our approach, similar to (Cogswell et al., 2016), an additional term is defined over the output maps of the layers. For a layer with $C$ neurons and a batch size of $N$, the additional computational cost is $O(C^2(N + 1))$ for direct variant and $O(C^3 + C^2 N))$ for both the determinant and log of the determinant variants.

## 3 GENERALIZATION ERROR ANALYSIS

In this section, we analyze how the proposed within-layer diversity regularizer affects the generalization error of a neural network. Generalization theory (Zhang et al., 2017; Kawaguchi et al., 2017) focuses on the relation between the empirical loss, as defined in equation 2, and the expected risk defined as follows:

$$L(f) = \mathbb{E}_{(\boldsymbol{x},y) \sim \mathcal{Q}}[l(f(\boldsymbol{x}), y)], \tag{5}$$

where $\mathcal{Q}$ is the underlying distribution of the dataset. Let $f^* = \arg\min_f L(f)$ be the expected risk minimizer and $\hat{f} = \arg\min_f \hat{L}(f)$ be the empirical risk minimizer. We are interested in the estimation error, i.e., $L(f^*) - L(\hat{f})$, defined as the gap in the loss between both minimizers (Barron, 1994). The estimation error represents how well an algorithm can learn. It usually depends on the complexity of the hypothesis class and the number of training samples (Barron, 1993; Zhai & Wang, 2018).

---

[1]This is defined only if $\boldsymbol{S}$ is positive definite. It can be shown that in our case $\boldsymbol{S}$ is positive semi-definite. Thus, in practice we use a regularized version $(\boldsymbol{S} + \epsilon \boldsymbol{I})$ to ensure the positive definiteness.

Several techniques have been used to quantify the estimation error, such as PAC learning (Hanneke, 2016; Arora et al., 2018), VC dimension (Sontag, 1998; Harvey et al., 2017; Bartlett et al., 2019), and the Rademacher complexity (Xie et al., 2015b; Zhai & Wang, 2018; Tang et al., 2020). The Rademacher complexity has been widely used as it usually leads to a tighter generalization error bound (Sokolic et al., 2016; Neyshabur et al., 2018; Golowich et al., 2018). The formal definition of the empirical Rademacher complexity is given as follows:

**Definition 3.1.** *(Bartlett & Mendelson, 2002) For a given dataset with N samples $\mathcal{D} = \{\boldsymbol{x}_i, y_i\}_{i=1}^N$ generated by a distribution $\mathcal{Q}$ and for a model space $\mathcal{F} : \mathcal{X} \to \mathbb{R}$ with a single dimensional output, the empirical Rademacher complexity $\mathcal{R}_N(\mathcal{F})$ of the set $\mathcal{F}$ is defined as follows:*

$$\mathcal{R}_N(\mathcal{F}) = \mathbb{E}_\sigma \left[ \sup_{f \in \mathcal{F}} \frac{1}{N} \sum_{i=1}^N \sigma_i f(\boldsymbol{x}_i) \right], \tag{6}$$

*where the Rademacher variables $\sigma = \{\sigma_1, \cdots, \sigma_N\}$ are independent uniform random variables in $\{-1, 1\}$.*

In this work, we analyse the estimation error bound of a neural network using the Rademacher complexity and we are interested in the effect of the within-layer diversity on the estimation error. In order to study this effect, inspired by (Xie et al., 2015b), we assume that with a high probability $\tau$, the distance between the output of each pair of neurons, $(\phi_n(\boldsymbol{x}) - \phi_m(\boldsymbol{x}))^2$, is lower bounded by $d_{min}$ for any input $\boldsymbol{x}$. Note that this condition can be expressed in terms of the similarity $s$ defined in equation 4: $s_{nm} \leq e^{(-\gamma d_{min})} = s_{min}$ for any two distinct neurons with the probability $\tau$. Our analysis starts with the following lemma:

**Lemma 3.2.** *(Xie et al., 2015b; Bartlett & Mendelson, 2002) With a probability of at least $1 - \delta$*

$$L(\hat{f}) - L(f^*) \leq 4\mathcal{R}_N(\mathcal{A}) + B\sqrt{\frac{2\log(2/\delta)}{N}} \tag{7}$$

*for $B \geq \sup_{\boldsymbol{x},y,f} |l(f(\boldsymbol{x}), y)|$, where $\mathcal{R}_N(\mathcal{A})$ is the Rademacher complexity of the loss set $\mathcal{A}$.*

It upper-bounds the estimation error using the Rademacher complexity defined over the loss set and $\sup_{x,y,f} |l(f(x), y)|$. Our analysis continues by seeking a tighter upper bound of this error and showing how the within-layer diversity, expressed with $d_{min}$, affects this upper bound. We start by deriving such an upper-bound for a simple network with one hidden layer trained for a regression task and then we extend it for a general multi-layer network and for different losses.

## 3.1 SINGLE HIDDEN-LAYER NETWORK

Here, we consider a simple neural network with one hidden-layer with $M$ neurons and one-dimensional output trained for a regression task. The full characterization of the setup can be summarized in the following assumptions:

**Assumptions 1.**

- *The activation function of the hidden layer, $\phi(t)$, is a $L_\phi$-Lipschitz continuous function.*

- *The input vector $\boldsymbol{x} \in \mathbb{R}^D$ satisfies $||\boldsymbol{x}||_2 \leq C_1$.*

- *The output scalar $y \in \mathbb{R}$ satisfies $|y| \leq C_2$.*

- *The weight matrix $\boldsymbol{W} = [\boldsymbol{w}_1, \boldsymbol{w}_2, \cdots, \boldsymbol{w}_M] \in \mathcal{R}^{D \times M}$ connecting the input to the hidden layer satisfies $||\boldsymbol{w}_m||_2 \leq C_3$.*

- *The weight vector $\boldsymbol{v} \in \mathbb{R}^M$ connecting the hidden-layer to the output neuron satisfies $||\boldsymbol{v}||_2 \leq C_4$.*

- *The hypothesis class is $\mathcal{F} = \{f | f(\boldsymbol{x}) = \sum_{m=1}^M v_m \phi_m(\boldsymbol{x}) = \sum_{m=1}^M v_m \phi(\boldsymbol{w}_m^T \boldsymbol{x})\}$.*

- *Loss function set is $\mathcal{A} = \{l | l(f(\boldsymbol{x}), y) = \frac{1}{2}|f(\boldsymbol{x}) - y|^2\}$.*

- *With a probability $\tau$, for $n \neq m$, $||\phi_n(\boldsymbol{x}) - \phi_m(\boldsymbol{x})||_2^2 = ||\phi(\boldsymbol{w}_n^T \boldsymbol{x}) - \phi(\boldsymbol{w}_m^T \boldsymbol{x})||_2^2 \geq d_{min}$.*

We recall the following two lemmas related to the estimation error and the Rademacher complexity:

**Lemma 3.3.** *(Bartlett & Mendelson, 2002) For $\mathcal{F} \in \mathbb{R}^{\mathcal{X}}$, assume that $g : \mathbb{R} \to \mathbb{R}$ is a $L_g$-Lipschitz continuous function and $\mathcal{A} = \{g \circ f : f \in \mathcal{F}\}$. Then we have*

$$\mathcal{R}_N(\mathcal{A}) \leq L_g \mathcal{R}_N(\mathcal{F}). \tag{8}$$

**Lemma 3.4.** *(Xie et al., 2015b) Under Assumptions 1, the Rademacher complexity $\mathcal{R}_N(\mathcal{F})$ of the hypothesis class $\mathcal{F} = \{f | f(\boldsymbol{x}) = \sum_{m=1}^{M} v_m \phi_m(\boldsymbol{x}) = \sum_{m=1}^{M} v_m \phi(\boldsymbol{w}_m^T \boldsymbol{x})\}$ can be upper-bounded as follows:*

$$\mathcal{R}_N(\mathcal{F}) \leq \frac{2L_\phi C_{134}\sqrt{M}}{\sqrt{N}} + \frac{C_4|\phi(0)|\sqrt{M}}{\sqrt{N}}, \tag{9}$$

*where $C_{134} = C_1 C_3 C_4$ and $\phi(0)$ is the output of the activation function at the origin.*

Lemma 3.4 provides an upper-bound of the Rademacher complexity for the hypothesis class. In order to find an upper-bound for our estimation error, we start by deriving an upper bound for $\sup_{\boldsymbol{x},f} |f(\boldsymbol{x})|$:

**Lemma 3.5.** *Under Assumptions 1, with a probability at least $\tau^Q$, we have*

$$\sup_{\boldsymbol{x},f} |f(\boldsymbol{x})| \leq \sqrt{\mathcal{J}}, \tag{10}$$

*where $Q$ is equal to the number of neuron pairs defined by $M$ neurons, i.e., $Q = \frac{M(M-1)}{2}$, and $\mathcal{J} = C_4^2 \big(MC_5^2 + M(M-1)(C_5^2 - d_{min}^2/2)\big)$ and $C_5 = L_\phi C_1 C_3 + \phi(0)$,*

The proof can be found in Appendix 7.1. Note that in Lemma 3.5, we have expressed the upper-bound of $\sup_{\boldsymbol{x},f} |f(\boldsymbol{x})|$ in terms of $d_{min}$. Using this bound, we can now find an upper-bound for $\sup_{\boldsymbol{x},f,y} |l(f(\boldsymbol{x}), y)|$ in the following lemma:

**Lemma 3.6.** *Under Assumptions 1, with a probability at least $\tau^Q$, we have*

$$\sup_{\boldsymbol{x},y,f} |l(f(\boldsymbol{x}), y)| \leq (\sqrt{\mathcal{J}} + C_2)^2. \tag{11}$$

The proof can be found in Appendix 7.2. The main goal is to analyze the estimation error bound of the neural network and to see how its upper-bound is linked to the diversity, expressed by $d_{min}$, of the different neurons. The main result is presented in Theorem 3.7.

**Theorem 3.7.** *Under Assumptions 1, with probability at least $\tau^Q(1 - \delta)$, we have*

$$L(\hat{f}) - L(f^*) \leq 8\Big(\sqrt{\mathcal{J}} + C_2\Big)\Big(2L_\phi C_{134} + C_4|\phi(0)|\Big)\frac{\sqrt{M}}{\sqrt{N}} + (\sqrt{\mathcal{J}} + C_2)^2 \sqrt{\frac{2\log(2/\delta)}{N}} \tag{12}$$

*where $C_{134} = C_1 C_3 C_4$, $\mathcal{J} = C_4^2\big(MC_5^2 + M(M-1)(C_5^2 - d_{min}^2/2)\big)$, and $C_5 = L_\phi C_1 C_3 + \phi(0)$.*

The proof can be found in Appendix 7.3. Theorem 3.7 provides an upper-bound for the estimation error. We note that it is a decreasing function of $d_{min}$. Thus, we say that a higher $d_{min}$, i.e., more diverse activations, yields a lower estimation error bound. In other words, by promoting the within-layer diversity, we can reduce the generalization error of neural networks. It should be also noted that our Theorem 3.7 has a similar form to Theorem 1 in (Xie et al., 2015b). However, the main difference is that Xie et al. analyse the estimation error with respect to the diversity of the weight vectors. Here, we consider the diversity between the outputs of the activations of the hidden neurons.

## 3.2 BINARY CLASSIFICATION

We now extend our analysis of the effect of the within-layer diversity on the generalization error in the case of a binary classification task, i.e., $y \in \{-1, 1\}$. The extensions of Theorem 3.7 in the case of a hinge loss and a logistic loss are presented in Theorems 3.8 and 3.9, respectively.

**Theorem 3.8.** *Using the hinge loss, we have with probability at least $\tau^Q(1 - \delta)$*

$$L(\hat{f}) - L(f^*) \leq 4\Big(2L_\phi C_{134} + C_4|\phi(0)|\Big)\frac{\sqrt{M}}{\sqrt{N}} + (1 + \sqrt{\mathcal{J}})\sqrt{\frac{2\log(2/\delta)}{N}} \tag{13}$$

*where $C_{134} = C_1 C_3 C_4$, $\mathcal{J} = C_4^2(MC_5^2 + M(M-1)(C_5^2 - d_{min}^2/2))$, and $C_5 = L_\phi C_1 C_3 + \phi(0)$.*

**Theorem 3.9.** *Using the logistic loss $l(f(x), y) = \log(1 + e^{-yf(x)})$, we have with probability at least $\tau^Q(1 - \delta)$*

$$L(\hat{f}) - L(f^*) \leq \frac{4}{1 + e^{\sqrt{-\mathcal{J}}}}\left(2L_\phi C_{134} + C_4|\phi(0)|\right)\frac{\sqrt{M}}{\sqrt{N}} + \log(1 + e^{\sqrt{\mathcal{J}}})\sqrt{\frac{2\log(2/\delta)}{N}} \quad (14)$$

*where $C_{134} = C_1 C_3 C_4$, $\mathcal{J} = C_4^2(MC_5^2 + M(M-1)(C_5^2 - d_{min}^2/2))$, and $C_5 = L_\phi C_1 C_3 + \phi(0)$.*

The proofs are similar to Lemmas 7 and 8 in (Xie et al., 2015b). As we can see, for the classification task, the error bounds of the estimation error for the hinge and logistic losses are decreasing with respect to $d_{min}$. Thus, employing a diversity strategy can improve the generalization also for the binary classification task.

## 3.3 MULTI-LAYER NETWORKS

Here, we extend our result for networks with P ($> 1$) hidden layers. We assume that the pair-wise distances between the activations within layer $p$ are lower-bounded by $d_{min}^p$ with a probability $\tau^p$. In this case, the hypothesis class can be defined recursively. In addition, we replace the fourth assumption in Assumptions 1 with: $||W^p||_\infty \leq C_3^p$ for every $W^p$, i.e., the weight matrix of the $p$-th layer. In this case, the main theorem is extended as follows:

**Theorem 3.10.** *With probability of at least $\prod_{p=0}^{P-1}(\tau^p)^{Q^p}(1 - \delta)$, we have*

$$L(\hat{f}) - L(f^*) \leq 8(\sqrt{\mathcal{J}} + C_2)\left(\frac{(2L_\phi)^P C_1 C_3^0}{\sqrt{N}}\prod_{p=0}^{P-1}\sqrt{M^p}C_3^p + \frac{|\phi(0)|}{\sqrt{N}}\sum_{p=0}^{P-1}(2L_\phi)^{P-1-p}\prod_{j=p}^{P-1}\sqrt{M^j}C_3^j\right)$$

$$+ \left(\sqrt{\mathcal{J}} + C_2\right)^2\sqrt{\frac{2\log(2/\delta)}{N}} \quad (15)$$

*where $Q^p$ is the number of neuron pairs in the $p^{th}$ layer, defined as $Q^p = \frac{M^p(M^p - 1)}{2}$, and $\mathcal{J}^P$ is defined recursively using the following identities: $\mathcal{J}^0 = C_3^0 C_1$ and $\mathcal{J}^p = M^p C^{p2}\left(M^{p2}(L_\phi C_3^{p-1}\mathcal{J}^{p-1} + \phi(0))^2 - M(M-1)\frac{d_{min}^p{}^2}{2}\right)$, for $p = 1, \ldots, P$.*

The proof can be found in Appendix 7.4. In Theorem 3.10, we see that $\mathcal{J}^P$ is decreasing with respect to $d_{min}^p$. Thus, we see that maximizing the within-layer diversity, we can reduce the estimation error of a multi-layer neural network.

## 3.4 MULTIPLE OUTPUTS

Finally, we consider the case of a neural network with a multi-dimensional output, i.e., $y \in R^D$. In this case, we can extend Theorem 3.7 by decomposing the problem into D smaller problems and deriving the global error bound as the sum of the small D bounds. This yields the following two theorems:

**Theorem 3.11.** *For a multivariate regression trained with the squared error, we have with probability at least $\tau^Q(1 - \delta)$,*

$$L(\hat{f}) - L(f^*) \leq 8D(\sqrt{\mathcal{J}} + C_2)\left(2L_\phi C_{134} + C_4|\phi(0)|\right)\frac{\sqrt{M}}{\sqrt{N}} + D(\sqrt{\mathcal{J}} + C_2)^2\sqrt{\frac{2\log(2/\delta)}{N}} \quad (16)$$

*where $C_{134} = C_1 C_3 C_4$, $\mathcal{J} = C_4^2(MC_5^2 + M(M-1)(C_5^2 - d_{min}^2/2))$ and $C_5 = L_\phi C_1 C_3 + \phi(0)$.*

**Theorem 3.12.** *For a multi-class classification task using the cross-entropy loss, we have with probability at least $\tau^Q(1 - \delta)$,*

$$L(\hat{f}) - L(f^*) \leq \frac{D(D-1)}{D - 1 + e^{-2\sqrt{\mathcal{J}}}}\left(2L_\phi C_{134} + C_4|\phi(0)|\right)\frac{\sqrt{M}}{\sqrt{N}} + \log\left(1 + (D-1)e^{2\sqrt{\mathcal{J}}}\right)\sqrt{\frac{2\log(2/\delta)}{N}}$$
$$(17)$$

*where $C_{134} = C_1 C_3 C_4$, $\mathcal{J} = C_4^2(MC_5^2 + M(M-1)(C_5^2 - d_{min}^2/2))$ and $C_5 = L_\phi C_1 C_3 + \phi(0)$.*

The proofs can be found in Appendix 7.5. Theorems 3.11 and 3.12 extend our result for the multi-dimensional regression and classification tasks, respectively. Both bounds are inversely proportional to the diversity factor $d_{min}$. We note that for the classification task, the upper-bound is exponentially decreasing with respect to $d_{min}$.

## 4 RELATED WORK

**Diversity promoting strategies** have been widely used in ensemble learning (Li et al., 2012; Yu et al., 2011), sampling (Derezinski et al., 2019; Bıyık et al., 2019; Gartrell et al., 2019), ranking (Yang et al.; Gan et al., 2020), and pruning by reducing redundancy (Kondo & Yamauchi, 2014; He et al., 2019; Singh et al., 2020; Lee et al., 2020). In the deep learning context, various approaches have used diversity as a direct regularizer on top of the weight parameters. Here, we present a brief overview of these regularizers. Based on the way diversity is defined, we can group these approaches into two categories. The first group considers the regularizers that are based on the pairwise dissimilarity of components, i.e., the overall set of weights are diverse if every pair of weights are dissimilar. Given the weight vectors $\{\boldsymbol{w}_m\}_{m=1}^M$, Yu et al. (2011) define the regularizer as $\sum_{mn}(1 - \theta_{mn})$, where $\theta_{mn}$ represents the cosine similarity between $\boldsymbol{w}_m$ and $\boldsymbol{w}_n$. Bao et al. (2013) proposed an incoherence score defined as $-\log\left(\frac{1}{M(M-1)}\sum_{mn}\beta|\theta_{mn}|^{\frac{1}{\beta}}\right)$, where $\beta$ is a positive hyperparameter. Xie et al. (2015a; 2016) used $\text{mean}(\theta_{mn}) - \text{var}(\theta_{mn})$ to regularize Boltzmann machines. They theoretically analyzed its effect on the generalization error bounds in (Xie et al., 2015b) and extended it to kernel space in (Xie et al., 2017a). The second group of regularizers considers a more globalist view of diversity. For example, in (Malkin & Bilmes, 2009; 2008; Xie et al., 2017b), a weight regularization based on the determinant of the weights covariance is proposed and based on determinantal point process in (Kulesza & Taskar, 2012; Kwok & Adams, 2012).

Unlike the aforementioned methods which promote diversity on the weight level and similar to our method, Cogswell et al. (2016) proposed to enforce dissimilarity on the feature map outputs, i.e., on the activations. To this end, they proposed an additional loss based on the pairwise covariance of the activation outputs. Their additional loss, $L_{Decov}$ is defined as the squared sum of the non-diagonal elements of the global covariance matrix $\boldsymbol{C}$:

$$L_{Decov} = \frac{1}{2}(||\boldsymbol{C}||_F^2 - ||\text{diag}(\boldsymbol{C})||_2^2),\tag{18}$$

where $||.||_F$ is the Frobenius norm. Their approach, Decov, yielded superior empirical performance; however, it lacks theoretical proof. In this paper, we closed this gap and we showed theoretically how employing a diversity strategy on the network activations can indeed decrease the estimation error bound and improve the generalization of the model. Besides, we proposed variants of our approach which consider a global view of diversity.

## 5 EXPERIMENTAL RESULTS

In this section, we present an empirical study of our approach in a regression context using Boston Housing price dataset (Dua & Graff, 2017) and in a classification context using CIFAR10 and CIFAR100 datasets (Krizhevsky et al., 2009). We denote as Vanilla the model trained with no diversity protocol and as Decov the approach proposed in (Cogswell et al., 2016).

### 5.1 REGRESSION

For regression, we use the Boston Housing price dataset (Dua & Graff, 2017). It has 404 training samples and 102 test samples with 13 attributes each. We hold the last 100 sample of training as a validation set for the hyper-parameter tuning. The loss weight, is chosen from $\{0.00001, 0.00005, 0.0001, 0.0005, 0.001, 0.005\}$ for both our approach and Decov (Cogswell et al., 2016). Parameter $\gamma$ in the radial basis function is chosen from $\{0.00001, 0.0001, 0.01, 0.1.1, 10, 100\}$. As a base model, we use a neural network composed of two fully connected hidden layers, each with 64 neurons. The additional loss is applied on top of both hidden layers.

We train for 80 epochs using stochastic gradient descent with a learning rate of $0.01$ and mean square error loss. For hyperparamrter tuning, we keep the model that perform best on the validation and use it in the test phase. We experiment with three different activation functions for the hidden layers: Sigmoid, Rectified Linear Units (ReLU) (Nair & Hinton, 2010), and LeakyReLU (Maas et al., 2013).

Table 1: Mean average error of different approaches on Boston Housing price dataset

|  | ReLU | Sigmoid | LeakyReLU |
|---|---|---|---|
| Vanilla | 2.97 | 3.16 | 2.85 |
| Decov | 2.77 | 2.99 | 2.80 |
| Ours (direct) | 2.72 | 2.97 | 2.82 |
| Ours (det) | 2.68 | 2.87 | 2.83 |
| Ours (logdet) | **2.64** | **2.83** | **2.77** |

Table 1 reports the results in terms of the mean average error for the different approaches over the Boston Housing price dataset. First, we note that employing a diversification strategy (ours and Decov) boosts the results compared to the Vanilla approach for all types of activations. The three variants of our approach, i.e., the within-layer approach, consistently outperform the Decov loss except for the LeakyReLU where the latter outperforms our direct variant. Table 1 shows that the logdet variant of our approach yields the best performance for all three activation types.

## 5.2 CLASSIFICATION

For classification, we evaluate the performance of our approach on CIFAR10 and CIFAR100 datasets (Krizhevsky et al., 2009). They contain 60,000 $32 \times 32$ images grouped into 10 and 100 distinct categories, respectively. We train on the 50,000 given training examples and test on the 10,000 specified test samples. We hold the last 10000 of the training set for validation. For the neural network model, we use an architecture composed of 3 convolutional layers. Each convolution layer is composed of 32 $3 \times 3$ filters followed by $2 \times 2$ max pooling. The flattened output of the convolutional layers is connected to a fully connected layer with 128 neurons and a softmax layer. The different additional losses, i.e., ours and Decov, are added only on top of the fully connected layer. The models are trained for 150 epochs using stochastic gradient decent with a learning rate of $0.01$ and categorical cross entropy loss. For hyper-paramters tuning, we keep the model that performs best on the validation set and use it in the test phase. We experiment with three different activation functions for the hidden layers: sigmoid, Rectified Linear Units (ReLU) (Nair & Hinton, 2010), and LeakyReLU (Maas et al., 2013). All reported results are average performance over 4 trials with the standard deviation indicated alongside.

Tables 2 and 3 report the test error rates of the different approaches for both datasets. Compared to the Vanilla network, our within-layer diversity strategies consistently improve the performance of the model. For the CIFAR10, the direct variant yields more than $0.72\%$ improvement for the ReLU and $2\%$ improvement for the sigmoid activation. For the LeakyReLU case, the determinant variant achieves the lowest error rate. This is in accordance with the results on CIFAR100. Here, we note that our proposed approach outperforms both the Vanilla and the Decov models, especially in the sigmoid case. Compared to the Vanilla approach, we note that the model training time cost on CIFAR100 increases by $9\%$ for the direct approach, by $36.1\%$ for the determinant variant, and by $36.2\%$ for the log of determinant variant.

Table 2: Test error rates on CIFAR10

|  | ReLU | Sigmoid | LeakyReLU |
|---|---|---|---|
| Vanilla | $32.04 \pm 0.57$ | $33.78 \pm 0.64$ | $30.99 \pm 0.27$ |
| Decov | $\mathbf{30.98 \pm 0.25}$ | $32.22 \pm 0.51$ | $30.70 \pm 0.35$ |
| Ours (direct) | $31.28 \pm 0.49$ | $\mathbf{31.69 \pm 0.51}$ | $30.86 \pm 0.75$ |
| Ours (det) | $31.28 \pm 0.60$ | $32.92 \pm 0.49$ | $30.93 \pm 0.44$ |
| Ours (logdet) | $31.26 \pm 0.41$ | $32.61 \pm 0.46$ | $\mathbf{30.70 \pm 0.25}$ |

## 6 CONCLUSIONS

In this paper, we proposed a new approach to encourage 'diversification' of the layer-wise feature map outputs in neural networks. The main motivation is that by promoting within-layer activation diversity, neurons within the same layer learn to capture mutually distinct patterns. We proposed an additional loss term that can be added on top of any fully-connected layer. This term complements

Table 3: Test error rates on CIFAR100

|  | ReLU | Sigmoid | LeakyReLU |
|---|---|---|---|
| Vanilla | $65.81 \pm 0.42$ | $78.52 \pm 0.40$ | $64.90 \pm 0.22$ |
| Decov | $65.26 \pm 0.21$ | $77.08 \pm 0.47$ | $64.57 \pm 0.23$ |
| Ours (direct) | $64.95 \pm 0.32$ | $\mathbf{76.91 \pm 0.91}$ | $64.85 \pm 0.23$ |
| Ours (det) | $\mathbf{64.90 \pm 0.40}$ | $77.79 \pm 0.29$ | $\mathbf{64.46 \pm 0.40}$ |
| Ours (logdet) | $64.95 \pm 0.17$ | $77.70 \pm 0.61$ | $64.49 \pm 0.19$ |

the traditional 'between-layer' feedback with an additional 'within-layer' feedback encouraging diversity of the activations. We theoretically proved that the proposed approach decreases the estimation error bound, and thus improves the generalization ability of neural networks. This analysis was further supported by experimental results showing that such a strategy can indeed improve the performance of neural networks in regression and classification tasks. Our future work includes extensive experimental analysis on the relationship between the distribution of the neurons output and generalization.

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

# 7 APPENDIX

In the following proofs, we use Lipschitz analysis. In particular, a function $f : \mathbb{A} \to \mathbb{R}$, $\mathbb{A} \subset \mathbb{R}^n$, is said to be $L$-Lipschitz, if there exist a constant $L \geq 0$, such that $|f(\boldsymbol{a}) - f(\boldsymbol{b})| \leq L||\boldsymbol{a} - \boldsymbol{b}||$ for every pair of points $\boldsymbol{a}, \boldsymbol{b} \in A$. Moreover:

- $\sup_{\boldsymbol{x} \in A} f \leq \sup(L||\boldsymbol{x}|| + f(0))$.
- if $f$ is continuous and differentiable, $L = \sup|f'(\boldsymbol{x})|$.

## 7.1 PROOF OF LEMMA 3.5

**Lemma 3.5.** *Under Assumptions 1, with a probability at least $\tau^Q$, we have*

$$\sup_{\boldsymbol{x}, f} |f(\boldsymbol{x})| \leq \sqrt{\mathcal{J}}, \tag{19}$$

*where $Q$ is equal to the number of neuron pairs defined by M neurons, i.e. $Q = \frac{M(M-1)}{2}$, and $\mathcal{J} = C_4^2\big(MC_5^2 + M(M-1)(C_5^2 - d_{min}^2/2)\big)$ and $C_5 = L_\phi C_1 C_3 + \phi(0)$.*

*Proof.*

$$f^2(\boldsymbol{x}) = \left(\sum_{m=1}^M v_m \phi_m(\boldsymbol{x})\right)^2 \leq \left(\sum_{m=1}^M ||\boldsymbol{v}||_\infty \phi_m(\boldsymbol{x})\right)^2 \leq ||\boldsymbol{v}||_\infty^2 \left(\sum_{m=1}^M \phi_m(\boldsymbol{x})\right)^2 \leq C_4^2 \left(\sum_{m=1}^M \phi_m(\boldsymbol{x})\right)^2$$

$$= C_4^2 \left(\sum_{m,n} \phi_m(\boldsymbol{x})\phi_n(\boldsymbol{x})\right) = C_4^2 \left(\sum_m \phi_m(\boldsymbol{x})^2 + \sum_{m \neq n} \phi_n(\boldsymbol{x})\phi_m(\boldsymbol{x})\right) \tag{20}$$

We have $\sup_{w,\boldsymbol{x}} \phi(\boldsymbol{x}) < \sup(L_\phi |\boldsymbol{w}^T \boldsymbol{x}| + \phi(0))$ because $\phi$ is $L_\phi$-Lipschitz. Thus, $||\phi||_\infty < L_\phi C_1 C_3 + \phi(0) = C_5$. For the first term in equation 20, we have $\sum_m \phi_m(\boldsymbol{x})^2 < M(L_\phi C_1 C_3 + \phi(0))^2 = MC_5^2$. The second term, using the identity $\phi_m(\boldsymbol{x})\phi_n(\boldsymbol{x}) = \frac{1}{2}\big(\phi_m(\boldsymbol{x})^2 + \phi_n(\boldsymbol{x})^2 - (\phi_m(\boldsymbol{x}) - \phi_n(\boldsymbol{x}))^2\big)$, can be rewritten as

$$\sum_{m \neq n} \phi_m(\boldsymbol{x})\phi_n(\boldsymbol{x}) = \frac{1}{2} \sum_{m \neq n} \phi_m(\boldsymbol{x})^2 + \phi_n(\boldsymbol{x})^2 - \Big(\phi_m(\boldsymbol{x}) - \phi_n(\boldsymbol{x})\Big)^2. \tag{21}$$

In addition, we have with a probability $\tau$, $||\phi_m(\boldsymbol{x}) - \phi_n(\boldsymbol{x})||_2 \geq d_{min}$ for $m \neq n$. Thus, we have with a probability at least $\tau^Q$:

$$\sum_{m \neq n} \phi_m(\boldsymbol{x})\phi_n(\boldsymbol{x}) \leq \frac{1}{2} \sum_{m \neq n} (2C_5^2 - d_{min}^2) = M(M-1)(C_5^2 - d_{min}^2/2). \tag{22}$$

Here $Q$ is equal to the number of neuron pairs defined by M neurons, i.e, $Q = \frac{M(M-1)}{2}$. By putting everything back to equation 20, we have with a probability $\tau^Q$,

$$f^2(\boldsymbol{x}) \leq C_4^2\Big(MC_5^2 + M(M-1)(C_5^2 - d_{min}^2/2)\Big) = \mathcal{J}. \tag{23}$$

Thus, with a probability $\tau^Q$,

$$\sup_{\boldsymbol{x}, f} |f(\boldsymbol{x})| \leq \sqrt{\sup_{\boldsymbol{x}, f} f(\boldsymbol{x})^2} \leq \sqrt{\mathcal{J}}. \tag{24}$$

$\square$

## 7.2 PROOF OF LEMMA 3.6

**Lemma 3.6.** *Under Assumptions 1, with a probability at least $\tau^Q$, we have*

$$\sup_{\boldsymbol{x}, y, f} |l(f(\boldsymbol{x}), y)| \leq (\sqrt{\mathcal{J}} + C_2)^2 \tag{25}$$

*Proof.* We have $\sup_{\boldsymbol{x}, y, f} |f(\boldsymbol{x}) - y| \leq 2\sup_{\boldsymbol{x}, y, f}(|f(\boldsymbol{x})| + |y|) = 2(\sqrt{\mathcal{J}} + C_2)$. Thus $sup_{x,y,f}|l(f(x), y)| \leq (\sqrt{\mathcal{J}} + C_2)^2$. $\square$

## 7.3 PROOF OF THEOREM 3.7

**Theorem 3.7.** *Under Assumptions 1, with probability at least $\tau^Q(1 - \delta)$, we have*

$$L(\hat{f}) - L(f^*) \leq 8\left(\sqrt{\mathcal{J}} + C_2\right)\left(2L_\phi C_{134} + C_4|\phi(0)|\right)\frac{\sqrt{M}}{\sqrt{N}} + (\sqrt{\mathcal{J}} + C_2)^2\sqrt{\frac{2\log(2/\delta)}{N}} \quad (26)$$

*where $C_{134} = C_1 C_3 C_4$, $\mathcal{J} = C_4^2\left(MC_5^2 + M(M-1)(C_5^2 - d_{min}^2/2)\right)$, and $C_5 = L_\phi C_1 C_3 + \phi(0)$.*

*Proof.* Given that $l(\cdot)$ is $K$-Lipschitz with a constant $K = sup_{\boldsymbol{x},y,f}|f(\boldsymbol{x}) - y| \leq 2(\sqrt{\mathcal{J}} + C_2)$, and using Lemma 3.3, we can show that $\mathcal{R}_N(\mathcal{A}) \leq K\mathcal{R}_N(\mathcal{F}) \leq 2(\sqrt{\mathcal{J}} + C_2)\mathcal{R}_N(\mathcal{F})$. For $\mathcal{R}_N(\mathcal{F})$, we use the bound found in Lemma 3.4. Using Lemmas 3.2 and 3.6 completes the proof. $\square$

## 7.4 PROOF OF THEOREM 3.10

**Theorem 3.10.** *Under Assumptions 1, with probability of at least $\prod_{p=0}^{P-1}(\tau^p)^{Q^p}(1 - \delta)$, we have*

$$\begin{aligned}L(\hat{f}) - L(f^*) \quad \leq \quad & 8(\sqrt{\mathcal{J}} + C_2)\left(\frac{(2L_\phi)^P C_1 C_3^0}{\sqrt{N}}\prod_{p=0}^{P-1}\sqrt{M^p}C_3^p + \frac{|\phi(0)|}{\sqrt{N}}\sum_{p=0}^{P-1}(2L_\phi)^{P-1-p}\prod_{j=p}^{P-1}\sqrt{M^j}C_3^j\right) \\ & + \left(\sqrt{\mathcal{J}} + C_2\right)^2\sqrt{\frac{2\log(2/\delta)}{N}}\end{aligned} \quad (27)$$

*where $Q^p$ is the number of neuron pairs in the $p^{th}$ layer, defined as $Q^p = \frac{M^p(M^p-1)}{2}$, and $\mathcal{J}^P$ is defined recursively using the following identities: $\mathcal{J}^0 = C_3^0 C_1$ and $\mathcal{J}^p = M^p C^{p2}\left(M^{p2}(L_\phi C_3^{p-1}\mathcal{J}^{p-1} + \phi(0))^2 - M(M-1)d_{min}^2/2\right)$, for $p = 1, \ldots, P$.*

*Proof.* Lemma 5 in (Xie et al., 2015b) provides an upper-bound for the hypothesis class. We denote by $\boldsymbol{v}^p$ denote the outputs of the $p^{th}$ hidden layer before applying the activation function:

$$\boldsymbol{v}^0 = [\boldsymbol{w}_1^{0T}\boldsymbol{x}, ...., \boldsymbol{w}_{M^0}^{0T}\boldsymbol{x}] \quad (28)$$

$$\boldsymbol{v}^p = [\sum_{j=1}^{M^{p-1}} w_{j,1}^p\phi(\boldsymbol{v}_j^{p-1}), ...., \sum_{j=1}^{M^{p-1}} w_{j,M^p}^p\phi(v_j^{p-1})] \quad (29)$$

$$\boldsymbol{v}^p = [\boldsymbol{w}_1^{pT}\boldsymbol{\phi}^p, ..., \boldsymbol{w}_{M^p}^{pT}\boldsymbol{\phi}^p], \quad (30)$$

where $\boldsymbol{\phi}^p = [\phi(v_1^{p-1}), \cdots, \phi(v_{M^{p-1}}^{p-1})]$. We have

$$||\boldsymbol{v}^p||_2^2 = \sum_{m=1}^{M^p}(\boldsymbol{w}_m^{pT}\boldsymbol{\phi}^p)^2 \quad (31)$$

and $\boldsymbol{w}_m^{pT}\boldsymbol{\phi}^p \leq C_3^p\sum_n\phi_n^p$. Thus,

$$||\boldsymbol{v}^p||_2^2 \leq \sum_{m=1}^{M^p}(C_3^p\sum_n\phi_n^p)^2 = M^p C_3^{p2}(\sum_n\phi_n^p)^2 = M^p C_3^{p2}\sum_{mn}\phi_m^p\phi_n^p. \quad (32)$$

We use the same decomposition trick of $\phi_m^p\phi_n^p$ as in the proof of Lemma 3.5. We need to bound $\sup_x\phi^p$:

$$\sup_x\phi^p < \sup(L_\phi|\boldsymbol{w}_j^{p-1T}\boldsymbol{v}^{p-1}| + \phi(0)) < L_\phi||\boldsymbol{W}^{p-1}||_\infty||\boldsymbol{v}^{p-1}||_2^2 + \phi(0). \quad (33)$$

Thus, we have

$$||\boldsymbol{v}^p||_2^2 \leq M^p C^{p2}\left(M^2(L_\phi C_3^{p-1}||\boldsymbol{v}^{p-1}||_2^2 + \phi(0))^2 - M(M-1)d_{min}^2/2\right) = \mathcal{J}^P. \quad (34)$$

We found a recursive bound for $||\boldsymbol{v}^p||_2^2$, we note that for $p = 0$, we have $||\boldsymbol{v}^0||_2^2 \leq ||W^0||_\infty C_1 \leq C_3^0 C_1 = \mathcal{J}^0$. Thus,

$$\sup_{\boldsymbol{x},f^P\in\mathcal{F}^P}|f(\boldsymbol{x})| = \sup_{\boldsymbol{x},f^P\in\mathcal{F}^P}|\boldsymbol{v}^P| \leq \sqrt{\mathcal{J}^P}. \quad (35)$$

$\square$

## 7.5 Proofs of Theorems 3.11 and 3.12

**Theorem 3.11.** *For a multivariate regression trained with the squared error, we have with probability at least $\tau^Q(1-\delta)$,*

$$L(\hat{f}) - L(f^*) \leq 8D(\sqrt{\mathcal{J}} + C_2)\Big(2L_\phi C_{134} + C_4|\phi(0)|\Big)\frac{\sqrt{M}}{\sqrt{N}} + D(\sqrt{\mathcal{J}} + C_2)^2\sqrt{\frac{2\log(2/\delta)}{N}} \quad (36)$$

*where $C_{134} = C_1 C_3 C_4$, $\mathcal{J} = C_4^2(MC_5^2 + M(M-1)(C_5^2 - d_{min}^2/2))$, and $C_5 = L_\phi C_1 C_3 + \phi(0)$.*

*Proof.* The squared loss $||f(\boldsymbol{x}) - \boldsymbol{y}||^2$ can be decomposed into D terms $(f(\boldsymbol{x})_k - y_k)^2$. Using Theorem 3.7, we can derive the bound for each term. $\quad\square$

**Theorem 3.12.** *For a multiclass classification task using the cross-entropy loss, we have with probability at least $\tau^Q(1-\delta)$,*

$$L(\hat{f}) - L(f^*) \leq \frac{D(D-1)}{D-1+e^{-2\sqrt{\mathcal{J}}}}\Big(2L_\phi C_{134} + C_4|\phi(0)|\Big)\frac{\sqrt{M}}{\sqrt{N}} + \log\Big(1 + (D-1)e^{2\sqrt{\mathcal{J}}}\Big)\sqrt{\frac{2\log(2/\delta)}{N}}$$
$$(37)$$

*where $C_{134} = C_1 C_3 C_4$, $\mathcal{J} = C_4^2(MC_5^2 + M(M-1)(C_5^2 - d_{min}^2/2))$, and $C_5 = L_\phi C_1 C_3 + \phi(0)$.*

*Proof.* Using Lemma 9 in (Xie et al., 2015b), we have $\sup_{f,\boldsymbol{x},\boldsymbol{y}} l = \log\big(1 + (D-1)e^{2\sqrt{\mathcal{J}}}\big)$ and $l$ is $\frac{D-1}{D-1+e^{-2\sqrt{\mathcal{J}}}}$-Lipschitz. Thus, using the decomposition property of the Rademacher complexity, we have

$$\mathcal{R}_n(\mathcal{A}) \leq \frac{D(D-1)}{D-1+e^{-2\sqrt{\mathcal{J}}}}\left(\frac{2L_\phi C_{134}\sqrt{M}}{\sqrt{N}} + \frac{C_4|\phi(0)|\sqrt{M}}{\sqrt{N}}\right). \quad (38)$$

$\quad\square$

