# OpenReview forum: "ON NEURAL NETWORK GENERALIZATION VIA PROMOTING WITHIN-LAYER ACTIVATION DIVERSITY"
_ICLR.cc/2021/Conference — Reject_

### Official Review · AnonReviewer1 · 2020-10-26
**fragile foundations**

**Rating:** 3
**Confidence:** 3

**Review:**

Strong point: the paper addresses an important problem.

Three main weaknesses, which justify the score:
•	The theoretical developments presented in the paper build on the Rademacher complexity, but ignore the conclusions drawn by Zhang et al. in Section 2.2 of their ICLR 2017 paper (Understanding deep learning requires rethinking generalization).
•	The theoretical developments build on the assumption that (i) there exists a lower bound, valid for any input, to the distance between the output of each pair of neurons, and (ii) the proposed diversity loss increases this lower bound. Those two assumptions are central to the theoretical developments, but are quite arguable. For example, a pair of neuron that is not activated by a sample, which is quite common, leads to a zero lower bound.
•	Experimental validation are not convincing. Only shallow networks are considered (2 or 3 layers), and the optimization strategy, including the grid search strategy for hyperparameters selection, is not described.

Minor issue: positioning with respect to related works is limited. For example, layer redundancy (which is the opposite of diversity) has been considered in the context of network pruning: https://openaccess.thecvf.com/content_CVPR_2019/papers/He_Filter_Pruning_via_Geometric_Median_for_Deep_Convolutional_Neural_Networks_CVPR_2019_paper.pdf

---

> ### Author Response · Authors · 2020-11-16
> **Response to AnonReviewer1 (1/2)**
>
> We thank the Reviewer for characterizing the problem tackled in our paper as important.
> The Reviewer challenged the foundations of our approach. While we agree that statistical-based theory clearly does not fully explain the generalization of neural networks, we believe that the motivation of our approach is indeed well-founded. Below we list a few related points:
>
> *A: The theoretical developments presented in the paper build on the Rademacher complexity, but ignore the conclusions drawn by Zhang et al. in Section 2.2 of their ICLR 2017 paper (Understanding deep learning requires rethinking generalization).*
>
> 1-As the Reviewer pointed out, in  (Zhang et al. 2017), it is argued that a deep learning network can learn any arbitrary function and thus its Rademacher complexity is expected to be close to one in practice. This claim is verified empirically: a deep learning network is able to achieve reasonable accuracy on CFAR10 with random labelling. It should be noted that this finding is not considered surprising. In fact,  the issue has been discussed widely  (see $https://openreview.net/forum?id=Sy8gdB9xx$). Moreover, the reasoning made in Zhang et al. (Rademacher equal to 1) relies on the assumptions that the norm of all functions in the hypothesis set H is bounded by 1, which is not always the case in neural network context. Thus, statistical-based theoretical analysis in general and Rademacher complexity in particular are still valid  tools for studying the generalisation of neural networks in general cases [1,2,3]. \
> In addition, we would like to draw the attention of the Reviewer to the fact that the models used in Zhang et al. to achieve a reasonable accuracy on random labelling are relatively large (Alexnet and Inception V3), while in our paper Theorems 3.7, 3.8, 3.9, 3.10, 3.11, and 3.12 are valid for any neural network configuration with any given Rademacher complexity. In fact, Theorem 3.7 and 3.8 consider only networks with a single hidden layer.
>
> 2- We note that we are not characterizing the generalization of the model using the Rademacher complexity of the model directly,  but we are using the Rademacher complexity of the loss hypothesis set to characterize the estimation error. This is different from the case argued in Zhang et al. (see the answer of Xiang Zhang in $https://openreview.net/forum?id=Sy8gdB9xx$ for more elaboration about this point). Note that the upper-bound in Lemma 3.2 has two terms and only the first term is depending on the Rademacher complexity. In the formulation of Theorems 3.7, 3.8, 3.9, 3.10, 3.11, and 3.12, it can be seen that actually both terms are inversely proportional to $d_{min}$. Thus, if we approximate the Rademacher complexity of the large network by 1 (as argued in Zhang et al.), the total upper-bound is still inversely proportional to $d_{min}$ as $d_{min}$ still appears in the second term and thus the finding of theorem remains valid.
>
> [1] Foster, Dylan J., et al. Hypothesis Set Stability and Generalization. Advances in Neural Information Processing Systems. 2019. \
> [2] Wei, Colin, et al. Regularization matters: Generalization and optimization of neural nets vs their induced kernel. Advances in Neural Information Processing Systems. 2019. \
> [3] Poggio, Tomaso, Andrzej Banburski, and Qianli Liao. "Theoretical issues in deep networks: Approximation, optimization and generalization." arXiv preprint arXiv:1908.09375(2019).

---

> > ### Author Response · Authors · 2020-11-16
> > **Response to AnonReviewer1 (2/2)**
> >
> > *B: The theoretical developments build on the assumption that (i) there exists a lower bound, valid for any input, to the distance between the output of each pair of neurons, and (ii) the proposed diversity loss increases this lower bound. Those two assumptions are central to the theoretical developments, but are quite arguable. For example, a pair of neuron that is not activated by a sample, which is quite common, leads to a zero lower bound.*
> >
> > *(i) there exists a lower bound, valid for any input, to the distance between the output of each pair of neurons.* \
> > We agree that the above assumption mentioned by the Reviewer is impractical especially if the intermediate layer has ReLU activation (for a given sample, there is a large chance that two neurons will have a zero activation and thus $d_{min}$ in practice would correspond to zero). However, we note that in our theoretical analysis, we consider a relaxed variant of this assumption by introducing the relative probability $\tau$. In fact, we formulate our claim as 'With a probability $\tau, $ the distance between the output of each pair of neurons, $( \phi_n(\textbf{x}) - \phi_m(\textbf{x}))^2 $, is lower bounded by $d_{min}$ for any input $\textbf{x}$'. Thus, by introducing $\tau$, the assumption becomes practical and the findings of the Theorems remain valid and practical.  \
> > *(ii) the proposed diversity loss increases this lower bound. For example, a pair of neuron that is not activated by a sample, which is quite common, leads to a zero lower bound.* \
> > 1- Note that empirically the additional loss term is computed per-batch and, thus, if two given pairs of neurons in the same layers are both not activated for all the samples within the batch then the additional loss term is (1/N)*N=1. This is the maximum value of the additional loss term that we are trying to minimize. Thus, during the training the model will be more keen on separating their activation to minimize the additional loss. \\
> > 2- We note that $d_{min}$, defined as the minimum distance between the pair of neurons with a given probability $\tau$, is indeed lower bounded by zero. Moreover, for the case $d_{min}=0$, the upper-bound of the estimation error (Theorems 3.7, 3.8, 3.9, 3.10, 3.11) is maximized and hence the generalization gap increases, i.e., we do not have a good generalization gap anymore, whereas maximizing this term yields a tighter upper-bound and a better generalization.
> > It should be noted that Theorems 3.7-3.11 only claim that  a larger minimum distance between the neurons activations within the same layer for an input $\textbf{x}$ with a high probability $\tau$ yields a tighter bound and thus might help improving the generalization ability.
> >
> > *C: Experimental validation are not convincing. Only shallow networks are considered (2 or 3 layers), and the optimization strategy, including the grid search strategy for hyper parameters selection, is not described.*
> >
> > As stated in the end of the first paragraph of Section 5.2, the hyperparameter tuning protocol used in the classification is the same protocol as in the regression Section 5.1. However, we agree with the Reviewer that this can be confusing. To this end, we describe the training protocol for the classification in Section 5.2 and provide more details about the training protocol: \
> > The models were trained for 150 epochs using stochastic gradient descent with a learning rate of $0.01$ and categorical cross entropy loss. For hyperparameter tuning, we kept the model that performs best on the validation set and use it in the test phase. We experiment with three different activation functions for the hidden layers: Sigmoid, Rectified Linear Units (ReLU), and LeakyReLU. Moreover, in the revised version of the paper, we now report average competitive results over 4 trials with the standard deviation indicated alongside.
> > The main focus of the paper is to provide some theoretical insights on how the generalization error is related to the neuron activation outputs and to show how employing a diversity strategy can help close the generalization gap.
> >
> > *Minor issue: positioning with respect to related works is limited. For example, layer redundancy (which is the opposite of diversity) has been considered in the context of network pruning: https://openaccess.thecvf.com/content_CVPR_2019/papers/He_Filter_Pruning_via_Geometric_Median_for_Deep_Convolutional_Neural_Networks_CVPR_2019_paper.pdf*
> >
> > Thank you for pointing this out. We followed the Reviewer's suggestion and updated the related work section by citing the following references on pruning using the redundancy concept: (Kondo & Yamauchi, 2014; (He et al., 2019); (Singh et al.,2020); (Lee et al., 2020).

---

> > > ### Comment · AnonReviewer1 · 2020-11-24
> > > **Still not convinced**
> > >
> > > Thank you for your reply.
> > > I however still remain skeptical regarding the theoretical developments.
> > > In particular, the way \tau is manipulated is not convincing.
> > > On page 2, \tau is defined by assuming that ‘with a high probability \tau; the distance between the output of each pair of neurons, is lower bounded by d_{min} for any input x’.
> > > In the appendix (Section 7.1), when deriving one of the key contribution of the paper, Eq. (22) refers to this definition of \tau to justify that, with a probability \tau, the sum of the products between the activations of all pairs of distinct neurons is upper bounded by a function that decreases with d_{min}^2. Actually, this directly results from the fact that the definition of \tau is used to state that, with a probability \tau, the mean (over all pairs of different neurons) of the squared distance between the neuron activations is lower bounded by d_{min}^2
> > > This statement is incorrect. Indeed, this is not because a distance lower bound d_min is valid with probability \tau on individual pairs of activations, that the squared sum of the distances over all the N distinct pairs of neurons is lower bounded by N times d_min^2, with probability \tau.

---

> > > > ### Author Response · Authors · 2020-11-24
> > > > **Response to AnonReviewer1**
> > > >
> > > > Thank you for the catching this! The assumption defined in the paper lower-bounds $(\phi_n(x)−\phi_m(x ))^2$ for a neuron-pair (m,n) by $d_{min}$ with a probability $\tau$ and in (Eq. 22) this lower-bound was erroneously used over the sum of different pairs. However, a given network topology, the number of neurons in each layer is fixed, so we can still use the assumption defined in the paper to bound the distance of a single neuron-pair (m,n) and reformulate the probability of the sum to reflect all neuron-pairs: As each neuron-pair verifies the assumption with a probability $\tau$, then with a probability at least $\tau^Q$, the sum is lower-bounded by $M(M-1)*d_{min}$, where M is the number of neurons and $Q = \frac{M(M-1)}{2}$ is the number of neuron-pairs for M neurons in the layer. We have now corrected the theorems and proofs in the revised paper using $\tau^Q$.

---

### Official Review · AnonReviewer2 · 2020-10-28
**Missing many potential comparison partners**

**Rating:** 5
**Confidence:** 4

**Review:**

The paper proposed three ways of diversifying outputs of neurons, and the analysis showed that the generalisation bound becomes tighter when the neurons become more diversified. It is an interesting finding, along with theoretical results and empirical results. Although, from a practical perspective, there are still many concerns.

It is clear that by increasing d_min, the generalisation bound gets tighter. However, it is also obvious that there are other factors that one can control to make the bound tighter, and regularising other factors might be simpler in terms of implementation and optimisation.

1. The constant C_4 in the upper bound of the weight vector connecting the hidden-layer to the output neuron. \sqrt(J) decays linearly with C_4, and the first term in the generalisation bound for regression tasks decays quadratically w.r.t. C_4. Compared with a linear decay w.r.t. d_min, C_4 seems to be a better option to regularise neural networks. In practice, one can empose an \ell_2 regularisation on the top linear layer to control the overall norm of the weight matrix so that C_4 is controlled.

2. The constant C_5 = L_\phi C_1 C_3 + \phi(0). As we can see in the generalisation bound for regression tasks, \sqrt(J) decays quadratically w.r.t. C_5, which is even faster than the decay rate w.r.t. C_4. To control C_5, one can choose an activation function that has a small L_\phi, or to control the weight vectors to the activation function to have a small norm C_3. Both of them can be done relatively easily compared to optimising pair-wise similarity.

Overall, I think there are other regularisations suggested by the bound that could be put into practice, which might also lead to good generalisation, and also simpler optimisation problem.

---

> ### Author Response · Authors · 2020-11-16
> **Response to AnonReviewer2**
>
> We thank the Reviewer for finding our approach interesting both theoretically and empirically. We hope to provide clarifications and additional results to address the highlighted issues. Please, let us know if there are further questions.
>
> *Other constants appearing in the theorems:*
>
> Indeed, the foundation developed in our paper (Theorems 3.7 to 3.12) can be used beyond the diversity strategy and many other novel (or old) approaches can be theoretically motivated or inspired using our theory to improve the generalization ability of neural networks.
> From this perspective, the theoretical contribution of our paper can be seen as generic and can motivate future works. However, we should note that the focus of the paper is more toward the within-layer activation diversification strategy, which to the best of our knowledge has never been theoretically studied before, as opposed to weight decay for example which has been theoretically motivated in [1] or dropout with has been theoretically motivated in [2].
> Note that weight decay in particular can be motivated using our paradigm: As it can be seen in Theorems 3.7-3.12 and as already pointed out by the Reviewer, the upper-bounds found are inversely proportional to $C_3$ and $C_4$, which control the norm of the weight matrices. We agree with the remark of the Reviewer that $C_4$ and $C_5$ yield a quadratic decay compared to the linear decay with respect to $d_{min}$. However, one also should note that both of these constants have a strict lower-bound of 0 (which corresponds to a weight matrix of zero), and thus minimizing them has a certain practical limit, whereas $d_{min}$ does not have any upper limit in the case of ReLU activation for example (ReLu does not have an upper-bound and thus one can always find a distribution of activations within the same layer such that the minimum distance between each two neurons is 'well' spread). It is worth mentioning that both techniques are not mutually exclusive and one can, for example, employ a weight decay over the weight matrices and a within-layer diversity strategy over the activations of the layers. Note that in this work, we do not compete with or claim superiority to weight decay.  Our aim is to motivate a new direction of research, where  activation diversity is the centre of interest and to provide some insights to how the generalization error is related to the neuron outputs.
>
> [1] Bartlett, Peter L. and Mendelson, Shahar. "Rademacher and Gaussian complexities: Risk bounds and structural results." Journal of Machine Learning Research 3.Nov (2002): 463-482. \
> [2] Wan, Li, et al. "Regularization of neural networks using dropconnect." International conference on machine learning. 2013.

---

### Official Review · AnonReviewer3 · 2020-10-29
**A neat extension encouraging layer output diversity with theoretical backing**

**Rating:** 5
**Confidence:** 4

**Review:**

This paper proposes adding regularization terms to encourage diversity of the layer outputs in order to improve the generalization performance. The proposed idea is an extension of Cogswell's work with different regularization terms. In addition, the authors performed detailed generalization analysis based on the Rademacher complexity. The appearance of the term related to the layer output diversity in the generalization bound provides theoretical support for the proposed idea.

The main weakness of this paper, in my humble opinion, is the lack of important details or rigor in the experiments presented. For example, the authors didn't mention how the hyperparameter selection was conducted, what optimizer (and its parameters) was used, how many runs per result and the confidence interval, whether any test was done to establish statistical significance, why state-of-the-art architecture was not used for the image classification tasks, etc. Without these important details and rigorous comparison, it's hard to have high confidence in the reproducibility of the results.

Details:
1) Intro section. The line of work in "double descent" shows that overparameterization doesn't necessarily lead to overfitting. For completeness, it'll be good to mention this line of work and qualify the claim on overfitting.
2) End of section 2. The authors claim that the proposed diversity term induces "within-layer" feedback. The regularization term is computed on the outputs of a layer, which do depend on the parameters of the lower layers. So when backpropagation happens, it will affect the parameters of the lower layers. Therefore, "within-layer" feedback doesn't sound accurate to me.
3) Section 3.1, last bullet point. Should $\tau$ be introduced here?  Otherwise, where does the $\tau$ later used in Lemma 3.5, Lemma 3.6 and Theorem 3.7 come from?
4) Section 5. The proposed regularization terms don't seem cheap to compute for large networks with wide layers. It'll be helpful to measure the training cost increase.

---

> ### Author Response · Authors · 2020-11-16
> **Response to AnonReviewer3**
>
> We thank the Reviewer for the positive feedback and finding our contribution neat and theoretically founded. Below we want to address some questions that we think are important to clarify first:
>
> *1-For example, the authors didn't mention how the hyperparameter selection was conducted, what optimizer (and its parameters) was used*
>
> As stated in the end of the first paragraph of Section 5.2, the hyperparameter tuning protocol used in the classification is the same protocol used in the regression Section 5.1. However, we agree with the Reviewer that this can be confusing and some details are missing. To this end, we described the training protocol in the classification, i.e., Section 5.2, and provided more details about the training protocol.
> The models were trained for 150 epochs using stochastic gradient descent with a learning rate of $0.01$ and categorical cross entropy loss. For hyperparameter tuning, we kept the model that performs best on the validation set and use it in the test phase. We experimented with three different activation functions for the hidden layers: Sigmoid, Rectified Linear Units (ReLU), and LeakyReLU. Moreover, in the revised version of the paper, we now report average results over 4 trials with the standard deviation indicated alongside.
>
>
> *2-Intro section. The line of work in "double descent" shows that overparameterization doesn't necessarily lead to overfitting. For completeness, it'll be good to mention this line of work and qualify the claim on overfitting.*
>
> We indeed missed several relevant citations to prior works on 'double descent' as you mention, and we thank you  for pointing them out to us. We have updated the Introduction of the paper to reflect the connections to prior work more explicitly. The following references were mentioned:  (Belkin et al., 2019); (Advani et al., 2020); (Nakkiran et al., 2020).
>
>
>
> *3-End of section 2. The authors claim that the proposed diversity term induces "within-layer" feedback. The regularization term is computed on the outputs of a layer, which do depend on the parameters of the lower layers. So when back-propagation happens, it will affect the parameters of the lower layers. Therefore, "within-layer" feedback doesn't sound accurate to me.*
>
> We completely agree with the Reviewer that the regularization term is computed on the outputs of a layer, which does depend on the parameters of the lower layers. Indeed, the error induced by this additional term is back-propagated to the earlier layers, similar to any regularization technique. Here, the term 'within-layer' feedback refers to two things: (i) This feedback is computed using neurons within the same layer. (ii) Using this additional loss term, each neuron within the same layer provides a 'direct' feedback to the adjacent neurons. In the standard back-propagation, neurons also provide feedback to their same layer: after the forward pass, the signal goes backward to the layer providing a feedback form all the layers. However, this can be seen as an indirect feedback, whereas the additional term introduced in our paper provides a direct one.
>
>
> *4-Section 3.1, last bullet point. Should $\tau$ be introduced here? Otherwise, where does the  later used in Lemma 3.5, Lemma 3.6 and Theorem 3.7 come from?*
>
> The Reviewer is right. $\tau$ should be re-introduced in the assumptions. Thank you for pointing this out. We modified the last bullet point accordingly.
>
>
> *5-Section 5. The proposed regularization terms don't seem cheap to compute for large networks with wide layers. It'll be helpful to measure the training cost increase.*
>
> Thank you for the suggestion. We agree with the reviewer that it will be helpful to provide computational complexity analysis of our approach. We updated the manuscript, we report the theoretical computational complexity of the three variants in Section 2:
> "For a layer with $C$ neurons and a batch size of $N$, the additional computational cost is O($C^2(N+1)$) for direct variant and O($C^3 + C^2N)$) for both the determinant and log of the determinant variants."
> We also report the time cost increase of the training in the experimental Section:
> "Compared to the vanilla approach, we note that the model training time cost  on CIFAR100 increases by $9%$% for the  direct approach, by $36.1%$% for the determinant variant, and by $ 36.2$% for the log of determinant variant."

---

### Official Review · AnonReviewer4 · 2020-10-29
**Well written paper, could use more ablation study**

**Rating:** 6
**Confidence:** 3

**Review:**

In this paper, the authors propose a technique to encourage the within-layer
activation diversity and therefore improve the model performance.
Specifically, they design a within-layer loss that add penalty to the similar
neurons. They also showed that encouraging the within-layer diversity will
help reduce the generalization error.

The paper is well-presented and authors provided enough intuition as well as
theoretical evidence why the diversity would help. Although I did not check
all the proofs, the results seem to be right.

The definition of within-layer diversity seems to be simply the
concentration of the values of each individual neuron. How does that affect
the distribution of the layer output on the unit ball? Will this lead to a
output similar to 'binarized' output?

The experiment seems insufficient to support the argument. Only very simple
neural networks on two toy examples are provided. More ablation study of the
neural/layer output distribution would help better understanding this issue.

Overall I think this paper provides some insights to how the generalization
error is related to the neuron outputs and vote for accept.

---

> ### Author Response · Authors · 2020-11-16
> **Response to AnonReviewer4**
>
> Thank you for finding our paper well-presented and intuitive.  Your comments are clear and show both the main strengths of the paper and  how it can be improved in the future.
>
> *1-The definition of within-layer diversity seems to be simply the concentration of the values of each individual neuron. How does that affect the distribution of the layer output on the unit ball? Will this lead to a output similar to 'binarized' output?.*
>
> The  within-layer diversity  can be defined as the global pair-wise distance between the neurons in the same layers. By promoting diversity, we encourage different neurons within the same layer to learn different features. As the Reviewer mentioned, the paper conducts a generic theoretical analysis (valid for any neural network) and shows that diversity can improve generalization.  We agree with the Reviewer that it would be interesting to study the distribution of the layer output on the unit ball and to develop links between our approach and the binarized neural network. However, we believe that the theory developed in this study contributes to opening the door for studying the relationship between the distribution of the neurons outputs and the generalization ability of the model in neural networks. The suggestions of the Reviewer are definitely interesting future research directions starting from the analysis provided in our paper.
>
>
> *2-The experiment seems insufficient to support the argument. Only very simple neural networks on two toy examples are provided. More ablation study of the neural/layer output distribution would help better understanding this issue.*
>
> We completely agree with the Reviewer that more extensive experimental analysis can provide better insights on the links neurons outputs and generalization. In (Cogswell et al., 2016), a related approach based on the neurons output was proposed and an intensive experimental analysis on the relation between layer output distribution and generalization  was provided. In this manuscript, the main focus was to provide theoretical insights on how activation diversity can help improve generalization and in the experiments, we focused on how our approach interact with the different activation functions. To provide more experimental insights on the matter, we now report average competitive results over 4 trials with the standard deviation indicated alongside.
>
> Since it is difficult to proceed with such extensive experimental analysis in the ICLR2021 rebuttal period, we added this problem to section 6. CONCLUSION AND DISCUSSION.

---

### Decision · Program_Chairs · 2021-01-07
**Final Decision**

**Decision:**

Reject

**Comment:**

The paper looks into generalization performance of NNs in supervised learning setting. The authors propose a regularizer to enhance neuron diversity in each layer(within-layer activation diversity) as a regularizer to improve generalization.  The proposed idea is an extension of Cogswell's work with different regularization terms. The appearance of the term related to the layer output diversity in the generalization bound provides theoretical support for the proposed idea.They use Radamacher complexity as a tool to show this and bound the estimation error.

pros.
-The paper looks into an interesting problem. Designing a regularizer to improve generalization performance of NNs is of huge importance.
-The paper is well-presented and clear.

cons.
-The main drawback of the paper is lack of proper comparison to other regularizers and showing the uniqueness/superiority of this regularizer and how it improves over existing methods either theoretically or with experiments. Without that the significance of this work is limited.
-The authors response to Reviewer 2's comment was not convincing enough. I encourage the authors to improve this in the next iteration of the paper.
- i suggest doing a better job in including the related work as also mentioned by the reviewer.
-The experiment section can use more explanation and details on choice of hyper parameters, etc
- showing performance improvement for a deep architecture would definitely  improve the paper. In the current version only 2 and 3 layer toy examples are shown.